# DRiPLOF: An RPL Extension for Multi-Interface Wireless Sensor Networks in Interference-Prone Environments

**DOI:** 10.3390/s22103906

**Published:** 2022-05-21

**Authors:** Robbe Elsas, Eli De Poorter, Jeroen Hoebeke

**Affiliations:** IDLab, Department of Information Technology, Ghent University—imec, 9052 Ghent, Belgium; eli.depoorter@ugent.be (E.D.P.); jeroen.hoebeke@ugent.be (J.H.)

**Keywords:** 6TiSCH, co-existence, end-to-end IP, IEEE 802.15.4, IIoT, multi-hop, multi-interface, RPL, wireless, WSN

## Abstract

The Routing Protocol for Low-power and Lossy Networks (RPL) is a popular routing layer protocol for multi-hop Wireless Sensor Networks (WSNs). However, typical RPL configurations are based on decade-old assumptions, leading to a mismatch with: (1) advances in wireless hardware; and (2) growing wireless contention. To soften the impact of external stressors (i.e., jamming and interference), we extended RPL to exploit the capabilities of modern multi-interfaced wireless devices. More specifically, our main contribution is the design, development, and evaluation of a novel RPL Objective Function (OF) which, through simulations, is compared to traditional single-interface approaches and a state-of-the-art multi-interface approach. We examine two scenarios, with and without the injection of jamming, respectively. Our proposed OF is shown to outperform, or otherwise perform similar to, all alternatives considered. In normal conditions, it auto-selects the best interface whilst incurring negligible protocol overhead. In our jamming simulations, it provides stable end-to-end delivery ratios exceeding 90%, whereas the closest alternative averages 65% and is considerably less stable. Given we have open-sourced our development codebase, our solution is an ideal candidate for adoption by RPL deployments that expect to suffer interference from competing technologies or are unable to select the best radio technology a priori.

## 1. Introduction

Wireless Sensor Networks (WSNs) are increasingly considered an essential means of gathering information in order to gain competitive advantages. Especially in industrial settings, where the cost of gathering data through retro-fitting sites and processes with traditional wired solutions is sky-high, the appeal of modern wireless alternatives is understandable. In this competitive landscape, the Internet Engineering Task-Force (IETF) defines a number of open protocol layers with the goal of providing easy integration into existing Internet Protocol (IP) infrastructure through end-to-end IP connectivity.

Although many attempts at a consolidated IETF stack have been made, most recently resulting in the Internet Protocol (IP) Version 6 (IPv6) over the Time-Slotted Channel Hopping (TSCH) mode of IEEE 802.15.4e [1] (6TiSCH) specification [2,3], the Routing Protocol for Low-power and Lossy Networks (LLNs) (RPL) [4] has been a constant throughout these efforts. However, this status quo no longer reflects the capabilities of protocol layers below the routing layer, nor does it answer the threats facing present-day wireless deployments. On the Physical layer (PHY) level, advances in semiconductor manufacturing have made radio chips smaller, cheaper, and more energy efficient than ever, leading to off-the-shelf devices with multiple radios (i.e., multi-interfaced devices) and/or radios that support multiple PHYs. Moreover, as explained by Tabaja and Cohen [5], distance-vector routing protocols such as RPL are susceptible to external stressors such as jamming (malicious) and interference (non-malicious) because of unfortunate properties of certain network repair mechanisms, which may leave the network in a state of convergence for extended periods of time. Given that cross-technology interference has quickly become a chief concern for anyone deploying WSNs, a concept called co-existence, it seems only appropriate to try and harness (at the network-layer level) the capabilities introduced by recent hardware advancements, and of multi-interfaced devices in particular, as a way to (amongst other things) mitigate the negative effects of external stressors on network performance. We shall refer to RPL routing solutions that enable switching between radios at runtime as multi-interface solutions. Single-interface solutions, on the other hand, imply that each RPL-enabled device has only one interface or, otherwise, uses the same radio for all communications.

We are not the first to recognize the mismatch between traditional RPL routing configurations and hardware capabilities. According to Vilajosana et al. [6] (p. 611): “Future research directions may investigate the management and coexistence of dual-band/dual-radio technologies and how the [RPL] management plane should address such capabilities”. In fact, others have come up with extensions to RPL that retain compliance with RFC 6550 [4], such that they may be used as a drop-in replacement for single-interface approaches in existing deployments. However, many of these multi-interface solutions are inconsiderate of their compatibility with adjacent protocol layers (i.e., they typically do not mention how they should operate in conjunction with IPv6 over Low-power Wireless Personal Area Network (6LoWPAN) Neighbor Discovery (ND) [7,8]), do not consider downward traffic flows (Section 2.3), or are otherwise incomplete in their specification. A more in-depth discussion of related work is provided in Section 3.

In an effort to address these shortcomings, we first identified (in literature) a promising multi-interface routing protocol extension in the form of an RPL Objective Function (OF) (Section 2.2) and, from it, derived a working implementation. From there, we designed our own OF and tested it against the aforementioned literature-derived multi-interface solution [9,10], as well as a common single-interface approach [11]. The performance of each solution was assessed in two scenarios (with and without jamming) with 10 simulation runs per solution per scenario (i.e., 20 runs per solution).

In summary, the main contributions of this paper are:The definition of a rule-set for a new RPL OF called the Dual-Radio interface routing Protocol for LLNs Objective Function (DRiPLOF), which enables true multi-interface operation and provides a good balance between reliability and end-to-end latency, as well as an implementation of DRiPLOF and of another multi-interface capable OF derived from the work of Lemercier et al. [9,10] (see Section 3).Modifications and additions to the Contiki-NG [12] operating system (see Section 5.1), which not only bring it closer to 100% compliance with relevant standards [4,11], but also enables tracking of interface-level statistics, making it easier to develop new OFs for RPL networks with multi-interfaced devices.Modifications and additions to the Cooja network simulator [13] and the Cooja radio driver for Contiki-NG, allowing anyone to simulate multi-interfaced nodes running a complete protocol stack of their own design.Radio drivers for certain physical platforms (e.g., the Zolertia ReMote [14]), such that a newly developed OF may be deployed across a network of multi-interfaced RPL nodes immediately after the simulation stage.

This paper first explains how RPL works in Section 2 to better understand the related work in Section 3. Next, it outlines a rule-set for a new RPL OF and describes how we evaluated its performance in Section 4 and Section 5, respectively. Simulation statistics, and their relevance towards better harnessing the capabilities of multi-interfaced devices, with a special focus on RPL’s resilience against external stressors (such as jamming), are presented in Section 6. Finally, we summarize our findings and discuss future work in Section 7.

## 2. Routing Protocol for LLNs

RPL is a distance-vector routing protocol. It establishes a tree-like routing topology in the form of a Destination-Oriented Directed Acyclic Graph (DODAG) converging at a root. Each RPL node tracks three logical sets of neighboring nodes (Figure 1): (1) the candidate neighbor set; (2) the parent set; and (3) the preferred parent, i.e., the default next hop towards the root.

Maintaining these sets requires information contained in DODAG Information Objects (DIOs) (Figure 2), destined to either: (1) the link-local all-RPL-nodes IPv6 multicast address (ff02::1a_16_); or (2) the source of a DODAG Information Solicitation (DIS). It contains (amongst others) four values used for topology identification and maintenance: the RPLInstanceID, DODAGID, DODAG Version Number, and the Rank advertised by the DIO source. RFC 6550 [4] (Section 3.1.2) provides more information.

While mandatory only for solicited DIOs, a DODAG Configuration option (Figure 3) is often included in every DIO. It contains (amongst others) two values used for rank computation/comparison: MaxRankIncrease and MinHopRankIncrease (see [4] (Section 6.7.6)). RFC 6550 notes the following:

“*This information is configured at the DODAG root and distributed throughout the DODAG with the DODAG Configuration option. Nodes other than the DODAG root MUST NOT modify this information.*”[4] (p. 52)

### 2.1. Node Rank

A node’s rank (=the distance vector) is “a scalar representation of the location or radius of a node within a DODAG Version” [4] (p. 20). Along an end-to-end path to the root, ranks must decrease monotonically. This enables routing loop avoidance [15] (Section 4.1). Loop detection (≠avoidance) is in turn facilitated by RPL Packet Information (RPI) contained in (nearly) all data packets [4] (Section 11.2).

### 2.2. Upward Topology Construction and Maintenance

A node wanting to join a DODAG has INFINITE_RANK. To become a parent, it must (re-)attach to the DODAG and obtain a new rank by selecting a preferred parent. A node is attached when its parent set is not empty. During (preferred) parent selection, a node computes the rank it will itself advertise in DIOs. This process is governed by an Objective Function (OF). The OF used in an RPL instance is uniquely identified by the Objective Code Point (OCP) field of the DODAG Configuration option. According to RFC 6550 [4] (p. 67), “the candidate neighbor set is a subset of the nodes that can be reached via link-local multicast. The selection of this set is implementation and OF dependent.” From this set, a node filters candidate parents, as prescribed by the OF. Nonetheless, universal rules apply:All candidate parents must belong to the same DODAG version as the node itself.Within a DODAG version, a node’s advertised rank must be greater than the rank advertised by any of its parent set members.

The OF’s main objectives are preferred parent selection and rank computation. Typically, these operations require a routing metric [16]. It is useful to distinguish between: (1) a metric that a node can infer from the link towards a neighbor (a link metric) or based on a property it possesses (a node metric); and (2) a metric reported by a neighbor, based on the upward path it provides. For example, Objective Function 0 (OF0) [17] is commonly used. With OF0, forming a parent set and, subsequently, selecting a preferred parent requires a node to know what rank it would obtain if it were to choose a given candidate neighbor as its preferred parent. That is, while observing universal rules concerning the DODAG version and advertised rank, the parent set contains two nodes for which the resulting rank would be lowest, i.e., one preferred parent and a backup feasible successor. The generic equation for rank computation (that is, with OF0) is given by (Equation 1).
(1)$$R\left(n\right)=R\left(c\right)+\Delta R_{c}\;,$$

where:
R(n)the rank of node *n* if it would choose node *c* as its preferred parent;R(c)the advertised rank of candidate neighbor *c*;$\Delta R_{c}$the rank increase associated with the direct path between nodes *c* and *n* (typically a routing metric).

A common OF0 alternative is the Minimum Rank with Hysteresis Objective Function (MRHOF) [11]. With MRHOF, parent set formation/preferred parent selection involves calculating the “path cost” through every neighbor, keeping a number of neighbors through which the path cost is lowest (whilst observing universal/MRHOF-specific rules) as parents and selecting the lowest-cost parent as preferred if the cost improvement over the current preferred parent exceeds a threshold (=hysteresis component). Then, after rank (≠path cost) computation for every parent, a node determines the rank (and possibly metric) it will advertise in future DIOs. RFC 6719 [11] provides a more in-depth explanation.

### 2.3. Downward Traffic

Besides the predominant Multi-Point to Point (MP2P) traffic flows, RPL also supports Point to Multi-Point (P2MP) traffic, i.e., flowing down the DODAG. For this purpose, Destination Advertisement Objects (DAOs) (Figure 4) are propagated up the DODAG. They contain (amongst others) the RPLInstanceID, copied from DIO messages.

RPL specifies two Modes Of Operation (MOPs) for P2MP traffic, one of which may be enabled at a time in an RPL Instance: (1) storing mode (stateful); and (2) non-storing mode (source-routed). To facilitate these MOPs, a DAO contains groupings of RPL Target- and Transit Information options [4] (Sections 6.7.7 and 6.7.8). Each contiguous set of Transit Information options pertains to the preceding (contiguous) set of RPL Target options. That is, RPL Target options represent the reachability of IPv6 addresses/prefixes/multicast groups, while Transit Information options specify attributes of the paths to the targets (i.e., reachable destinations) that immediately precede them. For the sake of simplicity, assume that the DAO parent set contains only a node’s preferred parent, DAOs are always unicast, and there is only one root disseminating a single subnet-wide prefix.

#### 2.3.1. Non-Storing Mode

In non-storing mode, a message destined to a node within the DODAG is first sent to the root, which attaches hop-by-hop routing information to it [18,19]. Downward routes are formed through recursive look-ups in the root’s source routing table. To build this table, nodes address DAO messages to the root’s global/unique-local unicast address. DAOs thus follow the default route resulting from upward topology construction, as will data traffic destined to other nodes before being routed down from the root. Each node appends the DAO with a Target option for each of the self-owned (routable) addresses via which it wishes to be reachable (conditions apply; see RFC 6550 [4] (Section 9.4)), followed by a single Transit Information option containing the global address of its single DAO parent (i.e., its preferred parent).

#### 2.3.2. Storing Mode

In storing mode, each node keeps routing table entries for its entire sub-DODAG. A message destined to a node within the DODAG travels upward along the default route until it reaches a common ancestor. From there, the message is routed downward by examining the routing table of consecutive hops. To build those routing tables, each node unicasts DAOs to its preferred parent using a link-local source and destination address. A node must attach to those DAOs a Target option for each of its self-owned addresses via which it wishes to be reachable together with all Target options it received from its children. A single Transit Information option mainly serves a maintenance purpose.

## 3. Related Work

The impact of interference on wireless networks is a well-studied problem, and interference mitigation strategies have been proposed for every IETF protocol layer. For example, the PHY configuration of IEEE 802.15.4 [20] links can be adapted based on link quality [21]. At the Medium Access Control (MAC) layer, TSCH (one of several IEEE 802.15.4 MAC protocols) may use, e.g., interference-aware scheduling algorithms [22] or channel blacklisting schemes [23]. At the routing layer, large amounts of (non-intentional) cross-technology interference (e.g., in the 2.4 GHz range) can cause RPL end-to-end reliability to drop below 10% [24], whilst periodic jamming may cause severe network instability [5]. Despite the numerous scientific papers focusing on RPL optimizations (for, e.g., mobility support, authentication and security, Quality of Service (QoS), etc.), few RPL optimizations focus on improving interference robustness in general, let alone robustness against jamming. One example proposes the use of back-up RPL parents that are least likely to be impacted by the same jammer [25].

However, the main drawback of the aforementioned interference mitigation strategies (≠routing solutions) is that they employ single-interface routing solutions, and hence do not cope well with wide-band jamming/cross-technology interference and/or unfavorable electromagnetic properties of the environment (e.g., multi-path fading), which may render an entire frequency band unusable. Since our primary goal is to fully exploit modern multi-radio platforms, this section gives an overview of routing-centered solutions for multi-modal (i.e., multi-interface/multi-PHY) RPL-based communication approaches (≠interference mitigation strategies).

Lemercier et al. [9,10] identified three approaches to multi-interface management: Multiple RPL Instances (MI), Parent Oriented (PO), and Interface Oriented (IO):The MI solution requires one RPL Instance per interface type. Since a node can only belong to one DODAG per instance, belonging to multiple DODAGs means joining multiple instances (each governed by an OF). The OFs can then be tailored to interface types. As such, the main advantage of the MI approach is that it can reuse a generic RPL implementation with a per-technology OF. However, to prevent loops, packets may only switch once between instances to a higher RPLInstanceID. Hence, the MI approach can handle only one link failure along a path and packet switching is uni-directional. In contrast, the proposed PO and IO solutions leverage multi-interfaced nodes in a single DODAG.The PO solution is based on OF0 (see Section 2.2). It defines $\Delta R_{c}$ as the average link metric to a neighbor calculated over all interfaces. If an interface is unavailable, its link metric defaults to a fixed maximum value during averaging. Thus, a neighbor incurs a penalty for each interface through which it is unavailable. Unlike OF0, the parent set size is unlimited. The parent yielding the lowest computed rank still becomes the preferred parent, except when there already is one, and one of its interfaces becomes unavailable. Then, metric averaging for the preferred parent is deferred by one OF call. This delay aims to prevent DODAG instability. Note that the interface providing the best link metric towards a neighbor becomes the default interface for (unicast) communication with said neighbor, i.e., it becomes the preferred interface. Preferred interface selection is transparent to RPL. Lemercier et al.’s [9,10] addressing architecture is rather vague. The authors simply stated that “the PO solution combines multiple links into a single virtual link” [9] (p. 2). To the best of our understanding (no code was available for further examination), this means that each node owns a single link-layer address and a single routable and link-local IPv6 address, regardless of how many interfaces it possesses. Assuming each node possesses the same set of interfaces, in order for this to work, each node may possess at most one interface of a given type. Hence, when receiving a message from a given neighbor, it can uniquely identify the neighbor’s originating interface. Furthermore, we assume that each node keeps a single neighbor cache, a single default router list, and a single routing table and that each node (conceptually) tracks a single candidate neighbor set and parent set, respectively. Nodes presumably advertise the same rank across all interfaces. Conveniently, with one link-layer address per node and a single neighbor cache, the PO solution can work with 6LoWPAN ND [7,8], as-is. In addition, both storing and non-storing mode (Section 2.3) would work since nodes are the addressable entities and the default router list and routing table are node-scope as well.The IO proposal views each interface as an independent entity. To this extent, a node keeps track of candidate neighbor tuples (nodeID, interfaceID) and, for each physical neighbor, considers only the tuple with the best link quality for addition to the parent set (based on DODAG version and advertised rank) and subsequent rank computation (Equation 1). Rank computation remains largely identical, only now, $\Delta R_{c}$ is defined as the link metric of the selected tuple. From the parent set, the neighbor tuple with the lowest computed rank becomes preferred. Note that, contrary to the PO solution, there is no stability mechanism preventing erratic preferred parent changes upon link failures. The absence of any sort of stability mechanism, combined with the fact that every interface is a separate neighbor, is shown to lead to DODAG instability.

Bezunartea et al. [26] proposed a dual-radio RPL solution similar to the PO approach [9,10]. Firstly, it also builds a single DODAG. Second, each physical neighbor can be addressed through both interfaces with the same IPv6 address(es). Third, a physical node advertises the same rank across both interfaces. The parent set is formed based on the DODAG version and advertised rank of candidate neighbors. However, in this case, two parent sets are formed, one for each interface. Unfortunately, the authors did not specify how the OF subsequently performs rank computation and selects the preferred parent, nor how preferred interfaces are selected. They did, however, describe useful guidelines for RPL management traffic, which we adapted slightly (see Section 4.1).

Balmau et al. [27] described a multi-interface RPL extension wherein every interface is an RPL neighbor and a separate addressable entity, maintaining its own neighbor cache. To our understanding, each physical node keeps a single routing table and advertises the same rank across all interfaces. Furthermore, each interface has a unique link-layer address and, from it, derives a link-local unicast address. Each physical node is presumably assigned one routable address, through which it may be reached across all interfaces. If so, that would require it to verify the uniqueness of its routable address on all interfaces before it may be used on any of them. If the routable address is not unique by definition, verifying uniqueness through Duplicate Address Detection (DAD) would only work with basic ND [28,29], and not with its 6LoWPAN-optimized form [7,8]. What is more, while (for downward traffic) this solution would work well in storing mode (see Section 2.3.2) if each physical node exclusively unicasts DAOs to its preferred parent (which is a neighboring interface), there is likely no standard-compliant method to support non-storing mode (see Section 2.3.1). In addition, Balmau et al.’s [27] proposal suffers a similar problem to Lemercier et al.’s [9,10] IO solution, i.e., changing the preferred interface to a neighboring node automatically requires a parent change.

More recently, Rady et al. [30] investigated the use of a single radio interface with multiple PHY configurations managed by the link layer [31] and argued that RPL should be multi-PHY aware as well in order to “improve the network energy footprint by selecting less costly PHYs [as parent] when possible” [30] (p. 84467). Because their work was based on the 6TiSCH stack [2,3], which uses TSCH [20] (i.e., a synchronized MAC protocol), the authors could support multiple PHYs per interface. The authors’ proposal is closely related to Lemercier et al.’s [9,10] IO approach. More specifically, every PHY configuration is considered a separate neighbor (physical nodes, and not interfaces nor PHYs, remain the addressable entities on both the link- and routing-layer). Note that multi-PHY support is very similar to single-PHY multi-interface support, the main difference being that multi-PHY support requires neighboring nodes to agree which PHY to use and when, thus implying the need for a synchronized MAC protocol.

Most of the scientific papers discussed above are limited in their description and do not provide an open implementation, making it difficult to identify specific details and/or compare the performance of different solutions. Table 1 provides an overview of the discussed related work and indicates how we differ from prior work.

## 4. The Dual-Radio Interface Routing Protocol for LLNs

This section discusses the design of DRiPLOF, i.e., our own RPL OF with support for multiple radio interfaces per device.

Firstly, exposing the cost of interfaces to RPL does not mean every interface has to be an RPL neighbor. For example, with Lemercier et al.’s [9] PO solution, the OF is also influenced by interface metrics, but instead penalizes neighbors (i.e., neighboring physical nodes) for being unavailable over an interface. As such, RPL optimizes for interface redundancy, while the link-layer can optimize for another goal through preferred interface selection. Thus, with PO, switching preferred interfaces to a neighboring node does not automatically require a parent change. The decoupling of neighbors from interfaces also means that limiting parent changes has less impact on a network’s ability to respond to local wireless distortions, which is a problem faced by Rady et al. [30] and Balmau et al. [27], whom, unlike Lemercier et al.’s [9] IO solution, do use a mechanism (similar to MRHOF [11]) to prevent excessive parent changes. These arguments make a strong case for PO, and so, we carried over the aforementioned principles from Lemercier et al.’s [9,10] Parent-Oriented Objective Function (POOF). However, POOF has several downsides (see Section 4.4), one of which is its inconsistent way of limiting parent changes. Therefore, we based our DRiPLOF on MRHOF instead of OF0 (see Section 4.3).

The following subsections describe the provisions related to addressing and control traffic (Section 4.1), as well as semantics related to interfaces and metrics (Section 4.2).

### 4.1. Provisions for Multi-Interfaced Operation

To prevent ambiguity, the most basic provisions for DRiPLOF are:Physical nodes, not their interfaces, are the RPL neighbors.Physical nodes are the addressable entities, meaning: (1) all IPv6 addresses are node-scope; and (2) each node has one link-layer address, common to all interfaces.Each node may have at most one interface of a given type, allowing it to uniquely identify from which of its neighbor’s interfaces a packet originated.Each node owns the same set of interfaces or is otherwise aware of the interfaces it should possess (through a mechanism that is out of this paper’s scope).

IPv6 ND is optional. The provisions for its use are laid out in Appendix A. If it is not used, some other mechanism is required to keep inferred metrics updated (Appendix B). In any case, RPL control traffic should also accommodate optimal multi-interfaced operation. Especially if ND is not supported, a proper rule-set is required to keep interface metrics updated with minimal overhead. As such, we propose to use the following rules for RPL management traffic (adapted from Bezunartea et al. [26]):Broadcast DIOs must be sent on all interfaces.Unicast DIOs, when sent in response to a DIS, must be sent on the same interface as the incoming DIS.Unicast DIOs used for freshness probing (Appendix B) must be sent on all interfaces.Otherwise, unicast DIOs must be sent via the preferred interface towards a neighbor.Unicast DAOs must be sent to DAO parents via the preferred interface towards those parents.DAO Acknowledgments (ACKs), which are always unicast in response to a unicast DAO, must be sent on the same interface as the incoming DAO.Broadcast DISs must be sent on all interfaces.Unicast DISs used for freshness probing (Appendix B) must be sent on all interfaces.Otherwise, unicast DISs must be sent via the preferred interface towards a neighbor.

Other RPL control messages are not considered (for now). RPL control messages sent over multiple interfaces are not duplicates, i.e., they have different MAC sequence numbers [20] (p. 100). Note that, there is often only one DAO parent, which is also the preferred parent as determined by the OF.

### 4.2. Inferred Metrics and Preferred Interface Selection

Consider the concept of virtual links. A virtual link towards a neighbor abstracts a nominal amount of physical links. A physical link to a neighbor is “available” when the neighbor can be reached over an interface of the given type. From a node’s perspective, there is no difference between a physical link and the interfaces between which it exists, since each node may possess at most one interface of a given type. Hence, the concepts of interfaces and physical links are interchangeable. Assuming the use of link metrics [16], a physical link metric is associated with every interface of a neighbor. This physical link metric is inferred from link-layer operations and is also called an “inferred metric”. Through a normalization process (see Section 4.3), the inferred metrics of all interfaces of a virtual link are combined into a virtual link metric or “normalized metric”. Note that the use of an additive routing metric [16] (p. 8), such as the Expected number of Transmissions (ETX) [16] (Section 4.3.2), is mandatory.

Some routing metrics, including ETX, account for unsuccessful transmission attempts. As such, a node cannot assume a physical link to be down simply because it did not receive a link-layer acknowledgment. Hence, we introduce the concept of a metric threshold. When the inferred metric of a physical link is worse than this threshold (or no metric is available), said link is considered down or “unavailable”.

Every time the inferred metric of one of a neighbor’s interfaces changes, the preferred interface for unicast communication towards said neighbor is (re-)selected. To be clear, preferred interface selection is performed for every neighbor. Moreover, it is transparent to RPL. Anyone could select the preferred interface based on criteria of his/her own choosing, as long as they are enforced consistently throughout the network. We went with the simplest approach, i.e., the interface with the best inferred metric becomes preferred. With ETX, this means that the interface with the lowest ETX is preferred for a given neighbor.

### 4.3. Parent Selection and Rank Computation

During an OF call, the path cost through every member of the candidate neighbor set is first computed by adding two components, that is: (1) the normalized metric towards a candidate neighbor (in 16-bit representation; when using ETX, this means the metric is multiplied by a factor of 128); and (2) the normalized metric reported by that candidate neighbor in the DAG Metric Container option of its DIOs, or (as in our case) the rank advertised by the candidate neighbor when using ETX.

The normalized metric is set to the inferred metric of the preferred interface towards the given neighbor, that is if all its interfaces are available. Otherwise, its normalized metric is increasingly skewed towards a large constant value for every unavailable interface. More specifically, metric normalization is a two-step process consisting of: (1) calculating a normalization weight based on the number of unavailable interfaces towards a neighbor, according to (Equation 2); and (2) calculating the actual normalized metric based on this weight, a constant scale multiplier, and the inferred metric of the preferred interface towards the neighbor, according to (Equation 3). Note that our solution can be applied to devices with any number of interfaces.
(2)$$W=min\left(I_{node}-VL,IL_{max}\right)/IL_{div}\;,$$
(3)$$M_{norm}=\left(W\times S\right)+\left(\left(1-W\right)\times M_{pref}\right)\;,$$

where:
*W*the virtual link metric normalization weight;$I_{node}$the number of interfaces per node;VLthe number of valid physical links towards a given neighbor;$IL_{max}$the max number of invalid physical links towards any neighbor, must be ≥0 and $<I_{node}$;$IL_{div}$a constant divider, must be $>IL_{max}$;$M_{norm}$the normalized metric of the virtual link towards a given neighbor;*S*a constant scale multiplier;$M_{pref}$inferred metric of the preferred physical link towards a given neighbor.

The rules for forming the parent set, determining the preferred parent, computing the rank to advertise, and determining which path cost to advertise (and how) are identical to the corresponding rules laid out for MRHOF by RFC 6719 [11]. Note that nodes must advertise the same rank across all their interfaces.

Figure 5 shows an example of DRiPLOF operation. It assumes that: (1) IPv6 ND is not used; (2) we are using freshness probes (Appendix B); and (3) we are using ETX. When the root starts a DODAG, it broadcasts DIOs, including a Prefix Information Option (PIO), over all interfaces. Node #1 receives a DIO over its green interface and immediately sets it as preferred towards the root. Since #1 has not sent anything to the root yet, its green interface defaults to ETX =3. However, as the blue physical link to the root is yet unknown, the normalized metric to the root is penalized (0.25×8+0.75×3=4.25). After preferred interface selection towards the root (which was mandatory because one of its inferred metrics “changed”), #1 joins the DODAG and picks the root as its preferred parent. Next, when a DIO arrives at #1 over the blue interface, its inferred metric also defaults to ETX =3, and the normalized metric is no longer penalized. As #1 sends data to the root over its green interface and freshness probes (Appendix B) over both interfaces, the inferred metrics are updated continuously. Then, at a certain point, the green interface of #1 is jammed. At first, this causes a preferred interface switch. However, eventually, it leads to #1’s green physical link (to the root) exceeding the metric threshold, meaning the normalized metric of the virtual link (to the root) is penalized.

### 4.4. Theoretical Comparison with POOF

While the idea behind POOF and DRiPLOF is similar, the execution is very different. This stems from the side-effects introduced by wireless interfaces. For example, consider a likely scenario wherein each node has one sub-GHz interface, which is typically reliable and longer range, but at a reduced data rate, and one 2.4 GHz interface, typically with a higher data rate, but shorter range and less reliable. As shown in Figure 6a, POOF tends to make long paths. Besides, preferred interface selection typically favors low-rate links. Long paths and slow links mean relatively high end-to-end latency.

Now, consider the same network for DRiPLOF (see Figure 6b) and say that $I_{node}=2$, $IL_{max}=1$, $IL_{div}=4$, and S=8 (which is also the ETX metric threshold). As preferred interface selection is identical for POOF and DRiPLOF, we still favor low-rate links. However, sometimes, a shorter path is more sensible, even if a virtual link provides less redundancy (because one of its physical links is down). After all, the longer a path, the less reliable it becomes, and in Figure 6, it is clear that the direct physical link between #2 and the root offers the best inferred metric anyway. The paths resulting from DRiPLOF (Figure 6b) reflect this trade-off between path length and link redundancy with the goal of providing similar reliability and reduced end-to-end latency.

To be clear, there is a trade-off here. Encouraging interface redundancy remains a valid idea, and so, if the physical link between #2 and the root has an ETX ≥ 2, the normalized metric of the corresponding virtual link becomes ≥ 3.5, and #2 shall take the path through #1 instead, even if the combined ETX (1+2=3) of that path suggests that it is less reliable. That is, it takes this path unless the root was already #2’s preferred parent prior to the ETX increase, in which case, hysteresis might prevent #2 from changing preferred parent. This addresses the network stability concerns expressed by Lemercier et al. [9,10] in a different way. Their PO proposal was based on OF0 [16], and thus requires a custom stability mechanism that is difficult to enforce consistently. More specifically, in certain situations, they defer metric averaging for the currently preferred parent by one OF call. However, when the OF is called, it varies between operating systems. Since DRiPLOF is based on MRHOF, which has a built-in stability mechanism that is not defined with respect to implementation-dependent events, we do not introduce such inconsistencies.

## 5. Implementation and Simulation Setup

Most solutions described in Section 3 were not implemented for actual hardware. In contrast, we implemented our solution to run in both a simulator, as well as on actual hardware. Section 5.1 describes our major contributions to the Operating System (OS) of choice, while Section 5.2 discusses the simulator. Section 5.3 details how we simulated a network to compare the performance of various solutions (see Section 6). Lastly, Section 5.4 lists all evaluation metrics used to represent the simulation data.

### 5.1. Multi-Interface Support for Contiki-NG

We chose Contiki-NG [12] as our OS because it supports RPL, allows for (future) integration into a 6TiSCH-compliant stack, and comes with a simulator that runs binaries compiled from source. Since Contiki-NG does not support multiple radio interfaces, many modifications to the codebase were made (>12,000 additions and <300 deletions). These modifications are provided open-source, allowing developers to use the code for future multi-interface research. For an in-depth look, we kindly refer you to Appendix B, the Appendix A, and our GitHub repository. In summary, our main contributions to Contiki-NG consisted of:A multi-interface radio driver for Zolertia Zoul-based [32] platforms, which serves as an abstraction of underlying radio drivers, as well as a reworked Cooja radio driver based on the same abstraction principles.A MAC layer adapted from the Contiki-NG Carrier-Sense Multiple Access (CSMA) MAC layer to accommodate the newly developed radio-driver semantics. We did not alter the original MAC protocol as such, but rather, enabled (near) independent MAC layer functionality for every interface.Source files for POOF and DRiPLOF.An extension to the Contiki-NG link stats module, which enables multi-interfaced operation. The link stats module is now also responsible for: (1) tracking inferred metrics/interface availability; (2) performing preferred interface selection; (3) performing metric normalization and presenting the result to an OF source file; and (4) tracking the freshness (≠availability) of interfaces.A major rework of Contiki-NG’s RPL-Classic routing layer such that it complies with RFC 6719 [11] (Section 3.3) when used with MRHOF and MRHOF-derived OFs (such as DRiPLOF).A new probing target selection algorithm and parent selection algorithm such that they now factor in the freshness of interfaces when deciding which neighbor to probe/select as a preferred parent. You can read more about probing and its relation to interface freshness tracking in Appendix B.

### 5.2. Modifying Cooja to Support Multi-Interfaced Simulations

For our simulations, we used Cooja [13], a Java-written network simulator. Each network node or Cooja “mote” executes a Contiki-NG binary through the Java Native Interface (JNI) [33]. As such, Cooja motes can manipulate certain variables, trigger all actions corresponding to one clock tick, and check memory for changes. Based on these interactions, each mote manages a set of mote interfaces, which together represent the network node to the rest of the simulator. At the Contiki-NG end, the variables on which Cooja relies are manipulated primarily by the Cooja radio driver. At the Cooja end, a radio medium defines the behavior of the traffic exchanged between radios by implementing a propagation model. A radio is just a mote interface. Thus, apart from modifying the Cooja radio driver, we: (1) wrote a new radio class and attached an instance of it to every mote; and (2) modified the UDGM and LogisticLoss radio mediums to work with dual-radio motes. For more details, we kindly refer you to the Appendix A or to our Github repository.

For the simulations described in Section 5.3, we opted to use the modified LogisticLoss medium, which is based on two assumptions. Firstly, it assumes that the probability of correctly receiving a single transmission over a direct path is related to the signal strength at the receiver by means of a logistic function (Equation 4).
(4)$$PDR\left(RSSI\right)=\frac{1}{1+e^{RSSI_{50\%}-RSSI}}\;,$$

where:
PDRthe packet delivery ratio[−]RSSIthe received signal strength indicator[dBm]$RSSI_{50\%}$the RSSI for which PDR=0.5[dBm]*e*Euler’s constant[−]

Second, it assumes path loss relates to distance (between transmitter and receiver) by means of the log-distance path loss model with log-normal shadowing [34] (pp. 138–140) as given by (Equation 5).
(5)$$PL\left(d\right)=PL_{d_{ref}}+10\times \alpha \times \log_{10}\left(d/d_{ref}\right)+X_{\sigma }\;,$$

where:
PL(d)the path loss at distance *d* from the transmitter[dBm]*d*a given distance from the transmitter[m]$PL_{d_{ref}}$the path loss at distance $d_{ref}$ from the transmitter[dBm]$d_{ref}$a close-in (far-field) reference distance from the transmitter[m]αan empirically determined path loss exponent[−]$X_{\sigma }$random variable with Gaussian distribution, zero-mean, and standard deviation σ[dB]σstandard deviation of Gaussian random variable[−]

It is common practice to calculate the path loss at a close-in reference distance [34] (pp. 108–109) from the transmitter $PL_{d_{ref}}$ according to (Equation 6), i.e., the free-space path loss model derived from the Friis transmission equation [34] (pp. 107–108).
(6)$$PL_{d_{ref}}=20\times log_{10}\left(4\pi \frac{f}{c}d_{ref}\right)-AG_{t}-AG_{r}\;,$$

where:
*f*operating frequency of the transmitter/receiver[Hz]*c*the speed of light assuming a vacuum[m/s]$AG_{t}$antenna gain of the transmitter[dBi]$AG_{r}$antenna gain of the receiver[dBi]

Since the random variable $X_{\sigma }$ has a zero-mean, we can use (Equation 5) to find the average path loss at the maximum transmission distance $d_{max}$, also known as the transmission range. In addition, the average path loss at this distance equates to the signal transmit power $P_{t}$ minus the receiver sensitivity $S_{r}$, resulting in (Equation 7).
(7)$$\begin{array}{cc} \bar{PL(d_{max})} & =P_{t}-S\\ \; & =PL_{d_{ref}}+10\times \alpha \times \log_{10}\left(d_{max}/d_{ref}\right)\;, \end{array}$$

where:
$\bar{PL(d_{max})}$the average path loss at distance $d_{max}$ from the transmitter[dBm]$d_{max}$the maximum transmitting range[m]$P_{t}$signal transmit power[dBm]$S_{r}$sensitivity of the receiver[dBm]

Finally, by putting (Equation 6) into (Equation 7) and isolating for the transmission range $d_{max}$, we obtain (Equation 8). Since we are trying to model an industrial environment, which is typically indoors, we went with a reference distance $d_{ref}=1$ m, a common practice when modeling indoor environments [34] (p. 109).
(8)$$d_{max}=d_{ref}\times 10^{\left(P_{t}-S+AG_{t}+AG_{r}-20\times log_{10}\left(4\pi \frac{f}{c}d_{ref}\right)\right)/\left(\alpha \times 10\right)}\;,$$

Table 2 lists all relevant constants used to configure the LogisticLoss radio medium during our simulations (see Section 5.3). Note that we loosely based the values for α and σ on values found in literature [34,35,36,37,38]. However, since “it is better to utilize parameter values accurately characterizing the path loss for the specific scenario” [38] (p. 23), these values are likely not a good match for your specific use-case, and you should empirically determine more appropriate values [34] (Chapter 4.9.2). Nonetheless, to assess the effectiveness of DRiPLOF, it suffices that α and σ create a scenario wherein nodes have one interface with a relatively bad transmission success rate towards their closest neighbors and another interface for which the opposite is true.

### 5.3. Description of Simulation Setup

To compare our multi-interface-capable OF with existing solutions, we simulated an RPL network arranged in an equidistant grid of 25 by 25 nodes, one of which is the root node, as depicted in Figure 7. The distance between nodes (i.e., 50 m) serves no purpose apart from making sure that nodes have two interfaces with different transmission success rates, that is in combination with the values chosen for α and σ.

The flow of data transmission events depicted in Figure 8 ensures that medium access (and any attempt thereto) is spread relatively evenly in time across neighboring nodes. The root node also sends data packets back in response.

For the evaluation, we simulated a network wherein all nodes: (1) are dual-interfaced and run DRiPLOF; (2) are dual-interfaced and run POOF; (3) have a single 2400 MHz high-rate/short-range interface and run MRHOF; and (4) have a single 868 MHz low-rate/long-range interface and run MRHOF.

Simulations were performed both without and with an external jammer. In the first batch of simulations, no jamming took place and the performance of four OF/interface type configurations was evaluated in ten runs of 20 min each (each with a different random seed). All simulations were then repeated for a scenario wherein a jamming node starts blasting a given frequency band with corrupted data packets for approximately one minute at the 12 min mark (the interfering transmissions do not trigger packet processing). The inter-node spacing, combined with the constants used to configure the path loss model (Table 2) and the routing metric/preferred interface selection criteria, means that both DRiPLOF and POOF tend to favor the 868 MHz long-range/low-rate radio interface. Thus, jamming this interface represents a worst-case scenario when gauging interference resilience. Naturally, the single-interface configurations are jammed on their respective interface types. In addition, the jammer was placed close to the root because nodes closest to the root generally have the largest sub-DODAG. This way, the jammer can disrupt the entire network by forcing a local repair [5].

This makes for a total of 80 simulation runs for both scenarios combined. Table 3 provides an overview of common simulation parameters across all different simulation configurations.

### 5.4. Evaluation Metrics

The data gathered through the simulations are represented by means of certain evaluation metrics, which are defined as follows:The **end-to-end latency** is the time between a node instructing its transport layer to transmit a packet and it being processed by the root’s transport layer. This metric is only available when: (1) a node is part of the DODAG; and (2) a packet reaches the root.The **per-node number of parent changes** is defined, separately for each non-root node, as the number of times a non-root node switches preferred parent. The act of detaching from or (re-)attaching to the DODAG also counts as a parent change. This metric is deceiving when nodes are often detached from the DODAG.The **per-source Packet Delivery Ratio (PDR)** is defined, separately for each non-root node, as the ratio of data packets that were processed by the root’s transport layer to the number of send timer ($T_{s}$) expiration events. This metric does take into account data packets that: (1) were not sent up the DODAG because a node was detached; or (2) got dropped.The **time spent as orphan** is the time a non-root node spends detached from the DODAG. We start counting from the moment the root starts a DODAG or from the moment the non-root node booted if it did so after DODAG creation.The **per-source number of retransmissions** is defined, separately for each non-root node, as the total number of times it had to retransmit unicast packets to any next-hop neighbor because it did not receive a link-layer acknowledgment.The **per-node transmit energy** is defined, separately for each node (including the root), as the energy spent transmitting over the medium. It is calculated from the clock ticks the radios spent in transmit mode (reported to the Contiki-NG Energest module), the operating frequency, and transmit power. Since Energest cannot distinguish between interfaces, we assumed they all operated with the same transmit power (0 dBm).The **time-weighted average number of hops** is defined, separately for each non-root node, as the average number of hops along the path to the root, adjusted for the portion of simulation time for which a given path had said length. A hop is defined as a router along the path to the destination, which is neither the originator nor the endpoint of an IP datagram.

## 6. Simulation Results and Discussion

This section compares the performance of DRiPLOF with: (1) single-interface solutions (MRHOF); and (2) a state-of-the-art multi-interface solution (POOF). To this end, the overhead of DRiPLOF is evaluated both in conditions free of cross-technology interference (see Section 6.1) and in the presence of a jammer (see Section 6.2).

**The most important takeaway is that DRiPLOF performs similar to the best alternative (single- or multi-interface) under normal circumstances, while it is superior when external stressors such as jamming are involved.** This makes DRiPLOF especially suitable for RPL deployments that suffer interference from competing technologies such as WiFi^™^ [39] (in the 2.4 GHz band) and LoRaWAN^™^ [40] (in sub-GHz bands).

### 6.1. DRiPLOF’s Performance without External Interference

Intuitively, any multi-interface solution should outperform a single-interface solution due to its added flexibility. However, the added flexibility of multiple interfaces can actually be a disadvantage when deployed in “normal” conditions (i.e., without cross-technology interference). For example, energy consumption may increase since control traffic needs to be distributed over multiple interfaces, resulting in additional overhead. As such, to be useful, any multi-interface solution should not perform (significantly) worse than a single-interface solution under normal conditions.

Based on the evaluation metrics presented in Figure 9, the following observations can be made:DRiPLOF exceeds, or otherwise matches, the performance of the state-of-the-art POOF [9,10] in all evaluation metrics. Although the per-source PDR (see Figure 9c) and the number of retransmissions (see Figure 9g) are comparable, DRiPLOF forms a more stable DODAG, as indicated by the number of parent changes (see Figure 9b). This can be explained by the fact that POOF is based on OF0 [16] and, thus, requires an additional stability mechanism, which we found to be inconsistent because it relies on infrequent OF calls (the occurrence of which varies between implementations).DRiPLOF outperforms the single-interface approach using MRHOF with high-rate/short-range links (hereafter called “high-rate MRHOF”) in all cases, except for end-to-end latency (Figure 9a) and per-node transmit energy (Figure 9e). While the higher transmit energy cost of DRiPL is partially caused by the additional control traffic required for multi-interface operation, it is mainly because with the high-rate MRHOF, nodes spend more time as orphans (which cannot send data packets to the root and, hence, consume less transmit energy) (Figure 9d). Likewise, the end-to-end latency is artificially kept low, since dropped packets and detached/orphaned nodes are not accounted for (since those cases would equate to infinite latency).DRiPLOF matches, or is otherwise close to, the performance of the single-interface approach using MRHOF [11] with low-rate/long-range links (hereafter called “low-rate MRHOF”). Since DRiPLOF is based on MRHOF and because, with DRiPLOF, nodes generally prefer the low-rate interfaces, both solutions form relatively short and stable paths (Figure 9b,h). Although the transmit energy consumption of DRiPLOF is slightly higher due to control traffic duplication over both interfaces, in practice, this effect is negligible compared to the overall energy consumption of the devices. For all practical purposes, DRiPLOF matches the performance of the best single-hop solution we tested (when there is no cross-technology interference).

As such, DRiPLOF not only outperforms POOF [9,10] (i.e., a state-of-the-art multi-interface solution), but also has negligible overhead compared to single-interface solutions. Moreover, unlike a single-interface solution with a badly selected radio technology (e.g., in our simulations, using 2.4 GHz instead of 868 MHz), DRiPLOF automatically corrects for this mistake and switches to the most optimal interface. Note that, there is a point at which the inter-node spacing would be just small enough (or the transmit power high enough) for the end-to-end delivery rate of high-rate MRHOF to be comparable to low-rate MRHOF, POOF, and DRiPLOF. At that point, the end-to-end latency and per-node transmit energy of high-rate MRHOF would be much lower than any of its competitors, yet the per-node number of retransmissions would still be much higher. In future work, we may hence base DRiPLOF’s preferred interface selection on more than lowest ETX such that it may also pick the high-rate interface even if its ETX is higher, that is if doing so would result in improved energy consumption/latency.

### 6.2. DRiPLOF’s Performance in the Face of External Stressors

Finally, this section discusses the resilience of DRiPLOF against external stressors, as it was made clear by Tabaja and Cohen [5] that RPL deals poorly with repeated local repairs caused by, e.g., jamming or external interference.

Based on the evaluation metrics presented in Figure 10, the following observations can be made:The improved stability of DRiPLOF over POOF is significant, as shown by the number of parent changes (Note that, similar to end-to-end latency, the per-node number of parent changes may be kept artificially low if nodes are orphaned often. Nonetheless, if the time spent as an orphan is higher for a given solution while that solution also has a higher number of parent changes, that is a bad sign.) (Figure 10b) and time spent as an orphan (Figure 10d). These metrics indicate that DRiPLOF pays a much lower convergence penalty than POOF. That is, although the stability mechanism of POOF already causes more parent changes than DRiPLOF in baseline conditions, the difference is more pronounced here. Moreover, where previously the time spent as an orphan was similar, nodes now spend much more time as an orphan with POOF. This shows that POOF’s lesser stability mechanism is not the only culprit, since that leads to more switching between actual parents, not to nodes emptying their parent set. This can be explained by the unavailable interface penalty being much more severe for POOF, which increases the chance of having to drop the preferred parent. This is most problematic for nodes close to the root, whom typically have small parent sets because of rank requirements. If those drop their preferred parent, there is a high chance of them needing to perform a local repair. Even worse, these nodes often have large sub-DODAGs, meaning a local repair has great potential for high convergence penalties [5]. To be fair, we could lower POOF’s unavailable interface penalty, however not by such a quantity that its convergence penalty becomes trivial.DRiPLOF now outperforms all other OF/interface configurations. Especially noteworthy is the per-source PDR (Figure 10c), which was comparable between DRiPLOF, low-rate MRHOF, and POOF under baseline conditions, but only remained relatively stable for DRiPLOF when jamming was involved. Because we are blocking all possible connections to the root in the single-interface case, there is no point in comparing network stability between DRiPLOF and low-/high-rate MRHOF, since a single-interface solution can really only wait for the jamming to end.As mentioned before, end-to-end latency (Figure 10a) is artificially kept low when delivery rates (Figure 10c) are low and/or nodes spend much time being orphaned (Figure 10d). As such, Figure 10a should not be used to compare DRiPLOF with POOF nor low-/high-rate MRHOF, but rather with the end-to-end latency of DRiPLOF in non-jamming circumstances (Figure 9a), as the delivery rate and time spent as an orphan are relatively comparable. Doing so reveals that DRiPLOF’s ability to quickly adapt to changing conditions without incurring a large convergence penalty does not come at the cost of increased end-to-end latency.

As such, DRiPLOF is able to deliver packets with relatively stable end-to-end delivery rates exceeding 90% when the root and the nodes closest to it are being jammed, compared to a median delivery rate of 65% and lower for a state-of-the-art multi-interface solution (POOF) and single-interface solutions (low- and high-rate MRHOF).

## 7. Conclusions and Future Work

Due to advances in semiconductor manufacturing, off-the-shelf devices with multiple radios have become common and affordable. At the same time, the most commonly used routing protocol in WSNs (i.e., RPL) is known to be susceptible to external stressors such as jamming and interference. To remedy this, we adapted RPL to exploit the capabilities of modern multi-interfaced devices. That is, we developed a new RPL OF called DRiPLOF, and compared its performance with single-interface solutions (low- and high-rate MRHOF) and a state-of-the-art multi-interface approach (POOF).

To analyze the performance of our solution, we simulated a network wherein devices have a long-range/low-throughput 868 MHz radio and a short-range/high-throughput 2.4 GHz radio. In absence of jamming, DRiPLOF had negligible overhead compared to a single-interface alternative with optimal radio selection, yet it can adapt when a different radio type is more appropriate. When jamming was introduced, only DRiPLOF could maintain a stable network. Hence, it significantly outperformed all competitors, realizing stable end-to-end delivery rates exceeding 90% compared to a median 65% for the considerably less stable closest alternative (POOF).

Overall, this makes DRiPLOF an excellent candidate for adoption by RPL networks: (1) expecting to suffer interference from, e.g., competing technologies such as WiFi^™^ [39] (in the 2.4 GHz band) and LoRaWAN^™^ [40] (in sub-GHz bands); and (2) that are to be deployed in unknown conditions, meaning you cannot select the best radio technology a priori.

Nonetheless, future research efforts may yet improve upon our solution. For example, we assumed all devices had the same set of interfaces. However, one could devise a mechanism through which all nodes know which interfaces/how many they should possess. This would allow, e.g., nodes with a less-than-nominal amount of interfaces to join the network as well. Like Rady et al. [30], we could also support multiple PHY configurations per interface instead of just multiple single-PHY interfaces. However, this requires a synchronized MAC protocol (e.g., TSCH) because a node would need to know when its neighbor can receive over a certain interface with a given PHY configuration. With one PHY configuration per interface, this was not necessary as long as all interfaces were always-on. Finally, it would be interesting to see if basing preferred interface selection on more than the “best metric” could result in, e.g., an equally stable, but even more energy-efficient network (in certain circumstances). This might be achieved by combining multiple metrics for every interface (presumably prior to preferred interface selection), possibly with some sort of metric-weighting system, which in turn may or may not use some form of machine learning.

## Figures and Tables

**Figure 1 sensors-22-03906-f001:**
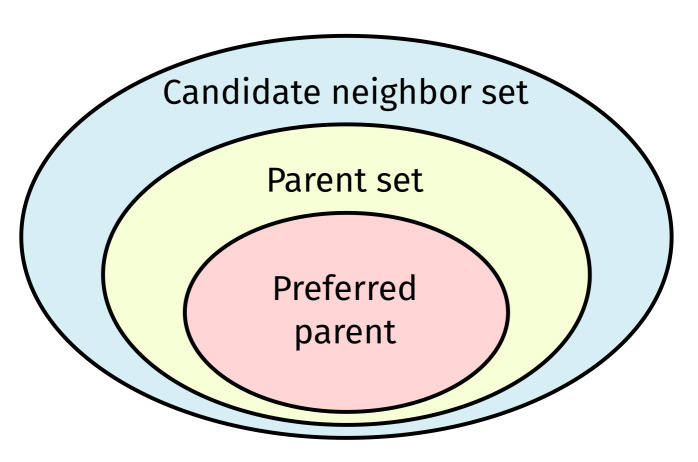
Relationship between three logical Routing Protocol for Low-power and Lossy Networks (LLNs) (RPL) sets tracked by each network node. Note that the formation of these sets is governed by the RPL Objective Function (OF) tied to the RPL instance to which a given node belongs.

**Figure 2 sensors-22-03906-f002:**
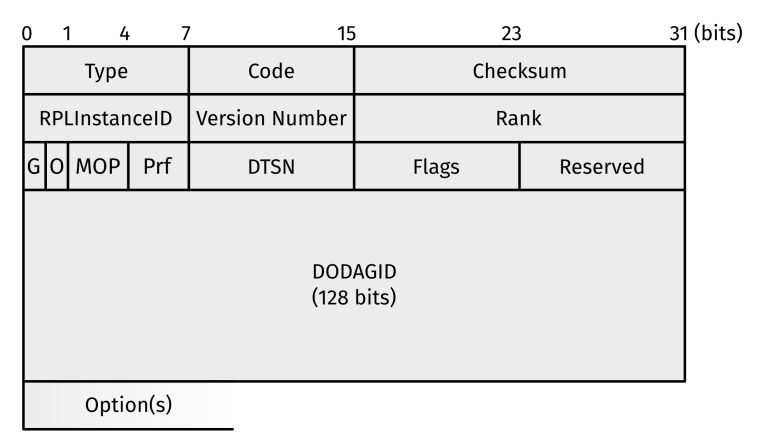
Format of an RPL Destination-Oriented Directed Acyclic Graph (DODAG) Information Object (DIO). Note that a DIO is an Internet Control Message Protocol (ICMP) Version 6 (ICMPv6) informational message with Type = 155 (RPL Control Message) and Code = 0x01 (DIO).

**Figure 3 sensors-22-03906-f003:**
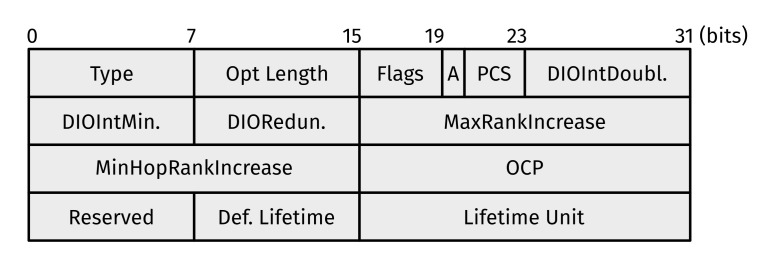
Format of a DODAG Configuration option. Note that a DODAG Configuration option is an RPL Control Message option with option Type = 0x04. In practice, it is only ever used to append DIO messages.

**Figure 4 sensors-22-03906-f004:**
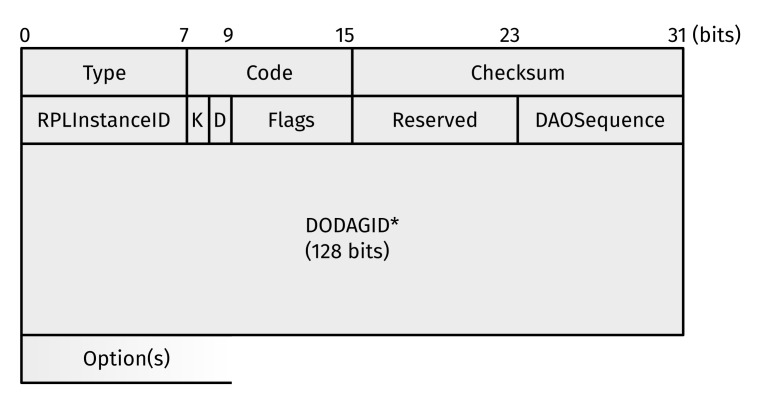
Format of an RPL Destination Advertisement Object (DAO). Note that a DAO is an ICMPv6 informational message with Type = 155 (RPL Control Message) and Code = 0x02 (DAO). The asterisk (*) indicates the absence of the DODAGID field in certain cases (see [4] (pp. 28–29)).

**Figure 5 sensors-22-03906-f005:**
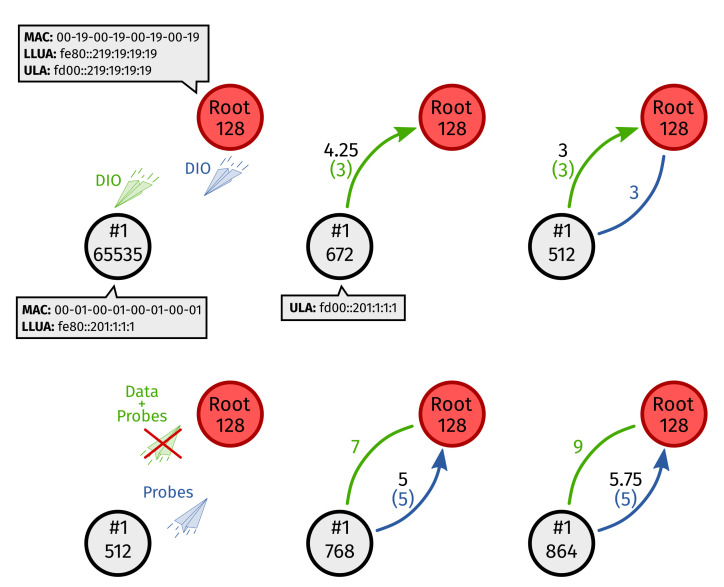
Operation example of the Dual-Radio interface RPL Objective Function (DRiPLOF). This figure should be viewed left to right and top to bottom. All nodes have a green and a blue interface. Inferred metrics are displayed in the same color as their corresponding interface; normalized metrics are written in black; the arrows indicate which interface is preferred. Note that $I_{node}=2$, $IL_{max}=1$, $IL_{div}=4$, and S=8.

**Figure 6 sensors-22-03906-f006:**
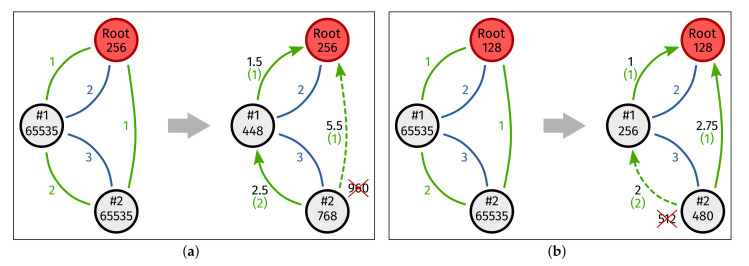
Comparison of upward topology formation between the Parent Oriented Objective Function (POOF) [9,10] and DRiPLOF. All nodes have a sub-GHz interface (green) and a 2.4 GHz interface (blue). Inferred metrics are displayed in the same color as their corresponding interface; normalized metrics are written in black; the arrows indicate which interface is preferred. (**a**) Upward topology formation under POOF. (**b**) Upward topology formation under DRiPLOF. Note that $I_{node}=2$, $IL_{max}=1$, $IL_{div}=4$, and S=8.

**Figure 7 sensors-22-03906-f007:**
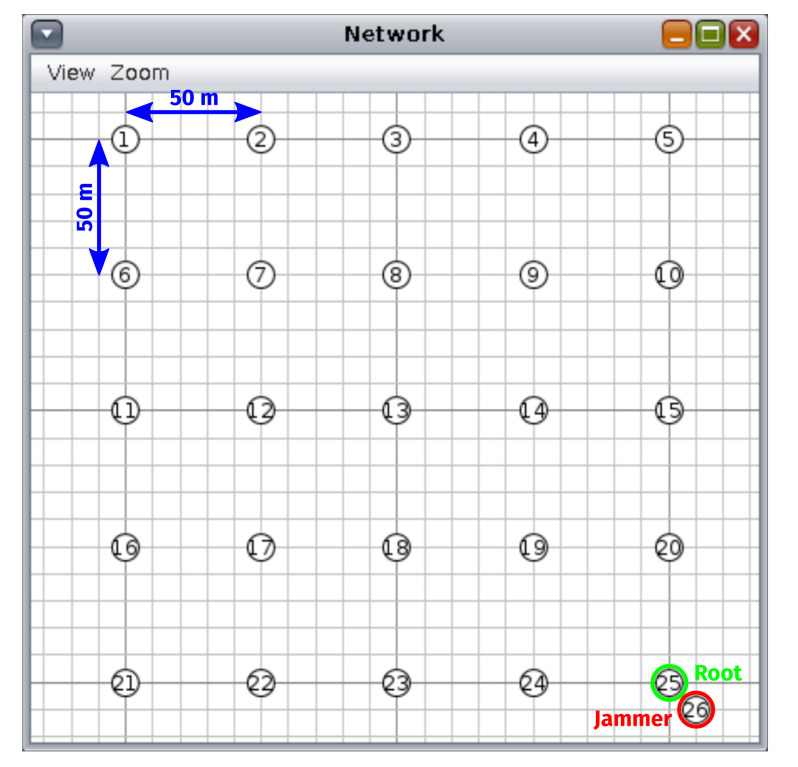
View of the simulated RPL network in the Cooja network window. Note that all simulated RPL nodes are arranged in an equidistant grid of 25 by 25 nodes. The RPL root is indicated in green, while the jammer (which is only active during half of the simulation runs) is indicated in red.

**Figure 8 sensors-22-03906-f008:**
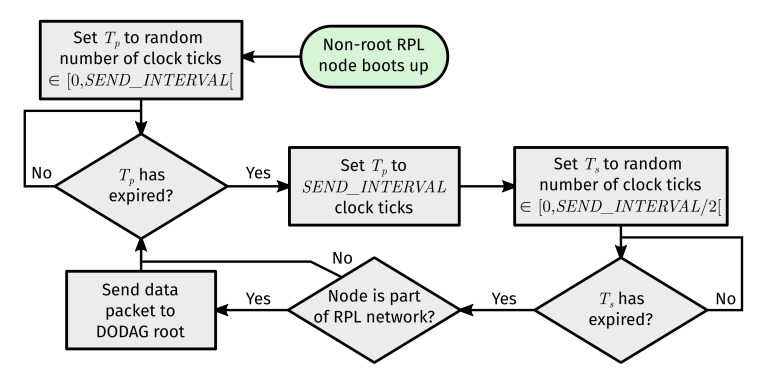
Flow of events related to pseudo-periodic data packet transmission (towards the root node) by non-root RPL nodes. Note that $T_{p}$ stands for “periodic timer”, $T_{s}$ stands for “send timer”, and SEND_INTERVAL is the amount of clock ticks in a time-span of ten seconds.

**Figure 9 sensors-22-03906-f009:**
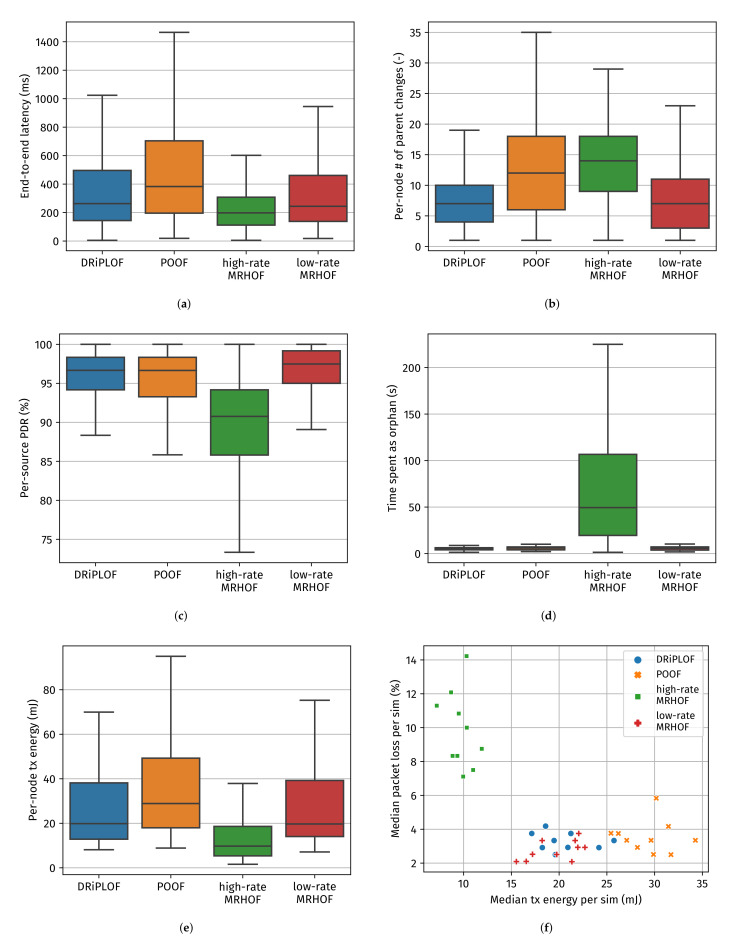
Simulation results for the conditions free of cross-technology interference. (**a**) The end-to-end latency of upward data packets successfully received by the root. (**b**) The number of times a non-root node changed its preferred parent. (**c**) The ratio of data packets processed by the root to the number of $T_{s}$ expiration events. (**d**) The time spent (by a non-root node) detached from the DODAG. (**e**) The energy spent (by a node) transmitting. (**f**) The median transmit energy versus the median packet loss for every simulation run. (**g**) The number of times a non-root node retransmitted a unicast packet. (**h**) Cumulative Distribution Function (CDF) of the time-weighted average path length.

**Figure 10 sensors-22-03906-f010:**
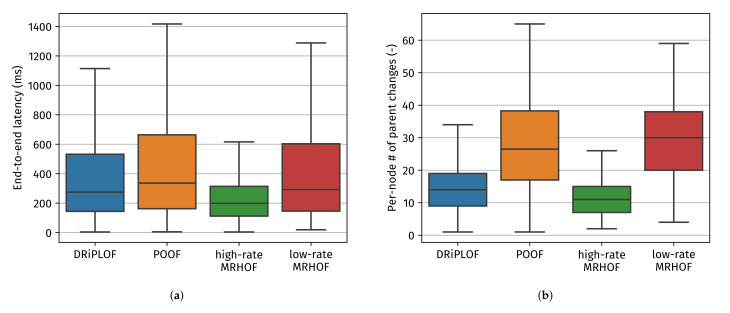
Simulation results for the conditions involving a jammer. (**a**) The end-to-end latency of upward data packets successfully received by the root. (**b**) The number of times a non-root node changed its preferred parent. (**c**) The ratio of data packets processed by the root to the number of $T_{s}$ expiration events. (**d**) The time spent (by a non-root node) detached from the DODAG.

**Table 1 sensors-22-03906-t001:** Comparison of IPv6 Routing Protocol for Low-Power and Lossy Networks (RPL) solutions that support multiple radio interfaces. Note that an ✗ indicates no support, a ✓ indicates full support, a ± sign signifies partial support, and a **?** means we did not have enough information to determine the kind of support (if any).

Reference	Solution Name	Single DODAG	Stability Mechanism	6LoWPAN ND Compatible ^a^	Works in Storing and Non-Storing MOP ^b^	Tested under Jamming	Firmware Available	Interfaces
[9,10]	MI	✗	✗	✓	✓	✗	✗	1 × 802.15.4g 1 × PLC
	PO	✓	±	✓	✓	✗	✗	
	IO	✓	✗	±	**?**	✗	✗	
[26]	—	✓	**?**	**?**	**?**	✗	✗	1 × 802.15.4 1 × 802.15.4g
[27]	—	✓	✓	✗	±	✗	✗ ^c^	1 × 802.15.4g 1 × PLC
[30]	—	✓	✓	±	±	✗	✓	2 × 802.15.4g
This paper	DRiPL	✓	✓	✓	✓	✓	✓	2 × 802.15.4 (any)

^a^ Supposed compatibility with Internet Protocol Version 6 (IPv6) over Low-power Wireless Personal Area Network (6LoWPAN) Neighbor Discovery (ND) is a purely theoretical assessment on our part; ^b^ support for a given Mode of Operation (MOP) is a purely theoretical assessment on our part; ^c^ although not publicly available, it is clear that the authors developed actual firmware.

**Table 2 sensors-22-03906-t002:** Overview of the constants used to configure the log-distance path loss model (with log-normal shadowing) in our simulations.

Constant	Interface #1	Interface #2
*f*	2400 MHz	868 MHz
$P_{t}$	0 dBm	0 dBm
*S*	−100 dBm	−100 dBm
$RSSI_{50\%}$ ^a^	−92 dBm	−92 dBm
$AG_{t}$	0 dBi	0 dBi
$AG_{r}$	0 dBi	0 dBi
*c*	$3.0\times 10^{8}$ m/s	$3.0\times 10^{8}$ m/s
α	3.0	3.0
σ ^b^	3.0	5.0
$d_{ref}$	1.0 m	1.0 m
$d_{\max}$	≈ 99.648 m	≈ 196.303 m

^a^*RSSI*_50%_ is only used for *PDR(RSSI)* calculations; ^b^ σ is only used for *PL(d)* calculations.

**Table 3 sensors-22-03906-t003:** Overview of common simulation parameters across different Objective Function (OF) and/or interface-type configurations.

Parameter	Value
Simulation duration	20 min/run
Simulation runs	10 runs/config (80 total)
Grid size	5 nodes × 5 nodes
Inter-node spacing	50 m
Data packet rate	≈6 packets/min
MAC protocol	CSMA-CA (always-on)
Data rate	low-rate: 25 kbps
	high-rate: 250 kbps

## Data Availability

The data presented in this study are available as Appendix A.

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
