# Peer review of "DRiPLOF: An RPL Extension for Multi-Interface Wireless Sensor Networks in Interference-Prone Environments"

_sensors, 2022, doi:10.3390/s22103906_

Round 1

Reviewer 1 Report

This manuscript is well-organized and with detailed experimental results. However, the introduction and related work part is too long, and it would be better to shorten these parts.

Reviewer 2 Report

  1. The description in the abstract section does not highlight the innovation of you work.
  2. Please describe the impact of your multi-interface scheme on adjacent layer protocols.
  3. How come the authors can have 5 contribution points.
  4. Authors intend to design full-feature routing protocol by considering more features. If the features are too many, how will you handle the weight of each feature? Is it possible to use machine learning? The following article is recommended to read and discuss.

“Novel Online Sequential Learning-based Adaptive Routing for Edge Software-Defined Vehicular Networks,” IEEE Transactions on Wireless Communications, 2020, DOI:10.1109/TWC.2020.3046275.

  1. Please summarize the difference between RPL objective function of yours and related work.
  2. Please explain the impact of interference on RPL, the advantages of your work on scenarios with interference needed to be highlighted.
  3. Please illustrate the source of the comparative experiments, which including the explanation of the compared methods and the references.
  4. Please conclude other solutions of multi-interface RPL dealing the impact of external stressors (i.e., jamming and interference).
  5. You may not number the bullet points by using 1,2,3, under a section like 4.3. It is not formal.

Reviewer 3 Report

The topic of this study is very interesting. The manuscript is well written. However, in order to improve the quality of the manuscript, my suggestions are given below.

1) In the introduction section please mention the benefits of this research, highlight your contribution using bullets or points and also mention the motivation of your study.

2) The introduction section is very brief, you should add a diagram and explain the concept of Multi-Interface.

3) Please add another comparison table in the related work section and compare the most recent studies.

4) in the simulation and experiment section please mention the simulation tool.

5)In the conclusion section please highlight the future research of this study.

Round 2

Reviewer 2 Report

  1. In the introduction, the last part of the second paragraph “We shall refer … for all communications.” is unclear.
  2. WSN is temporal. How to handle temporal feature of the network is a key. Please discuss this point and present how temporal feature is considered in your work. You can refer to the work. “A Novel Prediction-Based Temporal Graph Routing Algorithm for Software-Defined Vehicular Networks” IEEE Transactions on Intelligent Transportation Systems (T-ITS), 2021.
  3. In the second paragraph of Section 2, “ff02::1a16” should be explained what it means.
  4. Probably the quality of the pictures in figure 7 could be improved in the next version.
  5. The formats of tables should be unified. There is no table header in Table 2.
  6. MRHOF as the competitor for this paper should be presented in more detail in the previous section.
  7. The format 1.2… should be avoided when listing in the article in order to distinguish it from the paper title.
  8. In the figure of experimental results, different methods should preferably not be represented by different colors only.
  9. The conclusion part should be more refined to make the findings and contributions of the paper clearer. In addition, there should be no references in this section.
